# Interaction of the mitochondrial calcium/ proton exchanger TMBIM5 with MICU1

Li Zhang [1], Benjamin Gottschalk [2], Felicia Dietsche [1], Sara Bitar[1], Diones Bueno [1], Liliana Rojas-Charry[1], Anshu Kumari[3], Vivek Garg [3], Wolfgang F. Graier [2,4] & Axel Methner [1] ✉

Ion transport within mitochondria influences their structure, energy production, and cell death regulation. TMBIM5, a conserved calcium/proton exchanger in the inner mitochondrial membrane, contributes to mitochondrial structure, ATP synthesis, and apoptosis regulation. The relationship of TMBIM5 with the mitochondrial calcium uniporter complex formed by MCU, MICU1-3, and EMRE remains undefined. We generated *Tmbim5*-deficient Drosophila that exhibit disrupted cristae architecture, premature mitochondrial permeability transition pore opening, reduced calcium uptake, and mitochondrial swelling – resulting in impaired mobility and shortened lifespan. Crossing these with flies lacking mitochondrial calcium uniporter complex proteins was generally detrimental, but partial MICU1 depletion ameliorated the *Tmbim5*-deficiency phenotype. In human cells, MICU1 rescues morphological defects in TMBIM5-knockout mitochondria, while TMBIM5 overexpression exacerbates size reduction in MICU1-knockout mitochondria. Both proteins demonstrated opposing effects on submitochondrial localization and coexisted in the same macromolecular complex. Our findings establish a functional interplay between TMBIM5 and MICU1 in maintaining mitochondrial integrity, with implications for understanding calcium homeostasis mechanisms.

Mitochondria are intracellular organelles that harness a proton gradient generated by the electron transfer system in the inner mitochondrial membrane (IMM) to propel the $F_oF_1$-ATP synthase in a process called oxidative phosphorylation. The negative membrane potential also drives the passive uptake of $Ca^{2+}$ ions, which adapts mitochondrial ATP generation to demand by matching cellular and mitochondrial $Ca^{2+}$ levels during cellular activity and is important for cellular $Ca^{2+}$ signaling (reviewed by ref.[1]).

Mitochondrial $Ca^{2+}$ uptake predominantly occurs through the mitochondrial $Ca^{2+}$ uniporter (MCU), a pore-forming protein[2,3]. The MCU is the key component of the mitochondrial $Ca^{2+}$ uniporter complex (MCUC), which includes several interacting and regulatory proteins such as MICU1 (Mitochondrial Calcium Uptake 1)[4], MICU2[5], MICU3[6], the Essential MCU Regulator (EMRE)[7,8], and a dominant-negative subunit called MCUb[9]. Despite its critical role, the knockout (KO) of *Mcu* in mice exhibits strain-dependent effects, ranging from a mild skeletal muscle phenotype[10] to lethality during late embryonic development[11]. These observations suggest the presence of alternative mechanisms for $Ca^{2+}$ entry into mitochondria[12,13].

Recently, others and we proposed that the ubiquitously expressed protein Transmembrane Bax-Inhibitor Motif (TMBIM)-containing protein 5 (TMBIM5) represents an additional $Ca^{2+}$ and proton exchange system in the IMM. Austin et al.[14] and Patron et al.[15] both proposed that TMBIM5 works as an outward $Ca^{2+}$ transport system while we observed increased mitochondrial $Ca^{2+}$ uptake in cells overexpressing TMBIM5 stimulated with ER $Ca^{2+}$ release agents but no changes in mitochondrial release[16]. These results suggest that this new transport system can work in both directions depending on cellular demands. TMBIM5 is also part of the MCUC interactome as shown by proteomics[17], but the functional crosstalk between TMBIM5 and the MCUC is not well defined. This represents a significant gap in our understanding of the integrated networks regulating mitochondrial $Ca^{2+}$ homeostasis, cristae architecture, and tissue-specific vulnerabilities to mitochondrial dysfunction.

Mutation of the proposed channel pore of TMBIM5 in mice led to a massive downregulation of the TMBIM5 protein and caused a skeletal myopathy associated with swollen mitochondria and a disrupted cristae architecture[16]. However, it remained unclear whether this phenotype was caused by complete loss of TMBIM5 function or represented a dominant-negative effect of the mutant protein. Also in cells, *TMBIM5* deficiency leads to a disrupted cristae architecture[18,19] and an increased release of cytochrome *c* in response to apoptotic stimuli[19]. Additionally, TMBIM5 associates with

[1]University Medical Center of the Johannes Gutenberg-University Mainz, Institute for Molecular Medicine, Mainz, Germany. [2]Gottfried Schatz Research Center: Molecular Biology and Biochemistry, Medical University of Graz, Graz, Austria. [3]Department of Pharmacology and Physiology, School of Medicine, University of Maryland Baltimore, Baltimore, MA, USA. [4]BioTechMed, Graz, Austria. ✉e-mail: axel.methner@gmail.com

CHCHD2 (Coiled-Coil-Helix-Coiled-Coil-Helix Domain Containing 2), a protein that is part of the Mitochondrial Contact Site and Cristae Organizing System (MICOS) complex[20]. CHCHD2 deficiency results in similar effects as TMBIM5 deficiency, including altered cristae architecture and increased cytochrome *c* release[20]. This is of interest because mutations in the *CHCHD2* gene are linked to autosomal-dominant familial Parkinson's disease[21]. Together, these findings suggest that TMBIM5 is involved not only in Ca²⁺ transport but also in the stabilization of cristae architecture and the regulation of cytochrome *c* release.

Similar to *Tmbim5* loss of function in mice, mutations of *MICU1* in humans cause a proximal myopathy, additionally associated with learning difficulties and a progressive extrapyramidal movement disorder[22]. A detailed analysis of the mitochondrial Ca²⁺ uptake phenotype associated with disease-causing mutations and a skeletal muscle-specific knockout of *Micu1* in mice demonstrated a lower threshold for MCU-mediated Ca²⁺ uptake but also impaired mitochondrial Ca²⁺ uptake during excitation-contraction, aerobic metabolism impairment, muscle weakness, fatigue, and myofiber damage during physical activity[23]. Also similar to TMBIM5, MICU1 appears to play a role in the stability of cristae junctions and cytochrome *c* release[24] and interacts with CHCHD2[25]. Despite these parallel phenotypic consequences and shared interaction partners, a potential functional relationship between TMBIM5 and MICU1 remains inadequately characterized.

Besides a direct interaction also a more indirect interplay between TMBIM5 and the MCUC appears possible because TMBIM5 interacts with and inhibits AFG3L2[15]. AFG3L2 forms part of the m-AAA protease complex, which degrades misfolded or damaged proteins within the IMM. Importantly, the m-AAA protease also degrades non-assembled EMRE, the above-mentioned essential MCU regulator. This serves to ensure the efficient assembly of gatekeeper subunits with MCU and prevents the accumulation of constitutively active MCU-EMRE channels[26,27]. Changes in AFG3L2 activity imposed by changes in TMBIM5 abundance or interaction could, therefore, indirectly affect the composition of the MCUC. EMRE was not identified in the proteome analysis of TMBIM5-deficient cells reported by Patron et al.[15].

A recent comprehensive analysis of the phenotype of MCUC deficiency in *Drosophila melanogaster* strengthened the assumption that MICU1 has roles beyond regulating Ca²⁺ entry through the MCUC[28]. The fly genome encodes MCU, EMRE, MICU1, and MICU3, but lacks MICU2 and MCUb. Knockout of Mcu or Emre abolishes rapid mitochondrial Ca²⁺ uptake but results in only mild phenotypes and slightly shortened lifespans. In contrast, loss of Micu1 is developmentally lethal, and this cannot be rescued by simultaneous knockout of Mcu or Emre, indicating again uniporter-independent functions for Micu1, as also mentioned above. Mutants for Micu3 are viable but exhibit mild neurological impairments, and Micu1 and Micu3 are not functionally interchangeable[28].

In our study, we demonstrate that *Tmbim5* knockout in Drosophila closely resembles the findings obtained in mice, suggesting that the mouse phenotype is caused by loss of TMBIM5 function. Genetic interaction studies with the MCUC proteins conserved in flies revealed that concomitant *Micu1* heterozygosity and knockdown mitigated the *Tmbim5* loss-of-function phenotype, while concurrent knockout of the other MCUC components present in flies aggravated the phenotype. Further studies in human cells revealed a reciprocal regulatory relationship between MICU1 and TMBIM5 in mitochondrial structure maintenance. Our findings point to a functional interdependence between these proteins that maintains mitochondrial structural integrity, providing new insights into the molecular mechanisms governing Ca²⁺ homeostasis within mitochondria.

## Results

### TMBIM5 is upregulated in *MCU* and *MICU1* knockout cells
TMBIM5 is a Ca²⁺/H⁺ exchanger in the inner mitochondrial membrane[14–16] and interacts with the MCUC[17]. To investigate any potential compensatory changes of MCUC proteins in the absence of TMBIM5 hinting towards an involvement in common processes, we investigated the expression of MCU, MICU1, MICU2, and EMRE in *TMBIM5* knockout (KO) human embryonic kidney cells using immunoblotting. *TMBIM5* KO cells have a frameshift mutation (Fig. 1a) introduced by CRISPR/Cas9-mediated gene editing[16], resulting in loss of protein expression (Fig. 1b). We found a reduction in MCU abundance in these *TMBIM5* KO cells (Fig. 1c) but no changes in MICU1, MICU2, LETM1 (Fig. 1c, Supplementary data) or EMRE (Fig. 1d). The opposite experiment, studying TMBIM5 levels in the absence of MCUC proteins, demonstrated no apparent changes in *EMRE* KO cells (Fig. 1d) but an increased abundance of TMBIM5 in *MCU* and *MICU1* but not *MICU2* KO cells (Fig. 1e, Supplementary data). We interpreted these results as indicative of a potential crosstalk between TMBIM5 and components of the MCUC, specifically MCU and MICU1. The lack of reciprocal regulation between TMBIM5 and MICU2 likely reflects MICU2's functional dependence on MICU1, as MICU2 functions primarily as a MICU1 regulator within MICU1-MICU2 heterodimers rather than independently interacting with the MCU complex[5]. Importantly, EMRE levels were unchanged in *TMBIM5* KO and vice versa despite the inhibitory effect of TMBIM5 on the mAAA protease AFG3L2[15], which controls EMRE levels[26].

### Drosophila *Tmbim5* knockout impairs wing posture, mobility, and lifespan
To investigate genetic interactions between TMBIM5 and MCUC proteins, we next generated and characterized a *Tmbim5*-deficient *Drosophila melanogaster* model. This approach was chosen for its efficiency and the availability of well-characterized fly lines with deletions in key MCUC components[28]. The fly genome encodes orthologs of MCU, EMRE, MICU1, and MICU3, but lacks MICU2 and MCUb. However, unlike vertebrates, flies have two mitochondrial TMBIM family members, *CG2076* and *CG1287/Mics1*. We previously demonstrated that strong ubiquitous *CG2076* knockdown using the tubulin driver causes pupal lethality, while *CG1287/Mics1* knockdown leads only to male sterility, consistent with its testis-specific expression[29]. These findings suggested that *CG2076* is the likely ortholog of TMBIM5 in Drosophila. We, therefore, now generated a complete knockout of *Tmbim5* in w^1118 flies using CRISPR/Cas9 technology (Figure S1) to further study the effect of *Tmbim5* deficiency.

In contrast to RNAi-mediated knockdown of *Tmbim5*, *Tmbim5* KO flies were viable. Such differences are often observed and usually attributed to the fact that complete knockouts allow developmental adaptation and compensatory gene expression that can mitigate phenotypic severity[30]. Because *Tmbim5* is located on the sex chromosome, all male flies have a complete knockout of *Tmbim5* (Fig. 2a). Knockout males display a wing posture phenotype suggestive of flight muscle degeneration (Fig. 2b). This phenotype is often observed in flies lacking functional mitochondria[31,32]. *Tmbim5*-deficient flies also exhibit a severe reduction in their climbing ability (Fig. 2c) and lifespan (Fig. 2d), implying a strong effect of *Tmbim5* deficiency on the well-being of adult flies. Rescue experiments using the mild ubiquitous driver *daughterless*-Gal4 > UAS-*dTmbim5* confirmed that the lifespan defect was due to *Tmbim5* loss (Fig. 2d).

### Mitochondrial phenotype of *Tmbim5* knockout resembles mammalian models: altered cristae, mitochondrial swelling, and reduced Ca²⁺ buffering capacity
We then characterized the phenotype of these flies with a specific focus on the mitochondrial function. Already at day 7 after eclosion *Tmbim5* KO males have lower total ATP levels in the head and thorax (Fig. 2e). We chose this time point before the steep drop in lifespan to avoid the analysis of escapees. These lower ATP levels are not caused by a reduction in total mitochondrial mass demonstrated by unchanged mitochondrial DNA copy number (Fig. S2a) and citrate synthase activity (Fig. S2b). However, the cristae architecture in flight muscles was severely disrupted (Fig. 2f). While our transmission electron microscopy equipment limited detailed cristae structural analysis, previous studies have characterized these defects in detail, showing reduced cristae number, curved morphology, and ring-like inner membrane structures in *TMBIM5*-deficient cells[19]. In

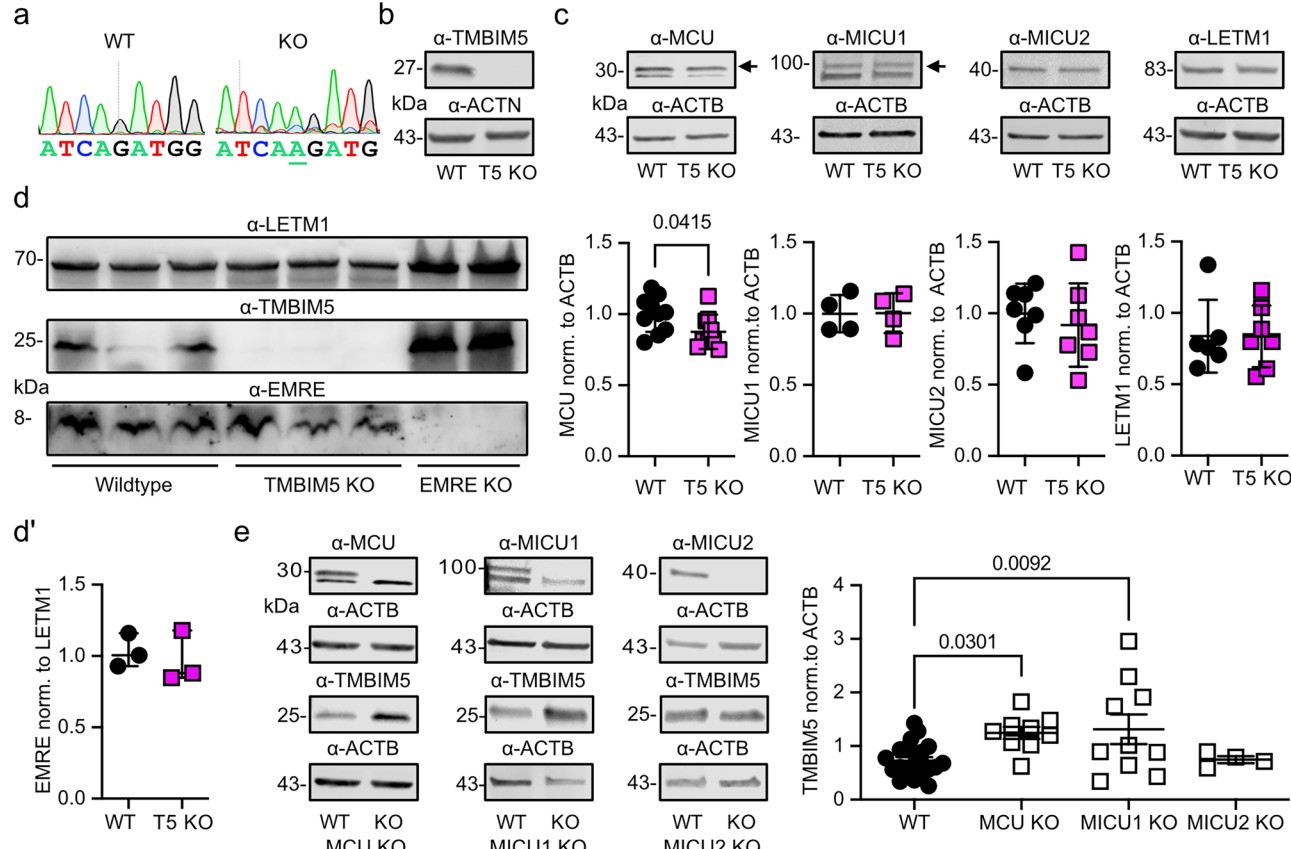

**Fig. 1 | TMBIM5 is upregulated in MCU and MICU1 knockout cells. a** DNA sequencing chromatograms of the genetic locus in wild type (WT) and TMBIM5-knockout (KO) HEK293 cells. The underlined additional adenosine results in a frameshift. **b** Immunoblot demonstrating loss of TMBIM5 protein expression in KO cells. Actin served as loading control, size is indicated. **c** Immunoblots demonstrating decreased levels of MCU (upper band, see (**e**)) and unchanged levels of MICU1, MICU2 and LETM1 in TMBIM5 (T5) KO cells. Actin served as loading control, size is indicated. **d** Immunoblot demonstrating preserved EMRE protein

levels in TMBIM5 KO cells. LETM1 served as loading control for mitochondrial mass and EMRE KO (loaded with higher protein concentration) cells to prove the specificity of the EMRE antibody. Size is indicated. **e** Immunoblots to demonstrate increased abundance of TMBIM5 in *MCU* and *MICU1* but not *MICU2* KO HEK293 cells. Actin served as loading control, size is indicated. Data in c and e are shown as mean ± SD, each data point represents an independent immunoblot. Statistical significance was calculated by the student's *t* test in (**c**) and one-way ANOVA followed by Holm-Šídák's multiple comparisons test in (**e**), *p* values are indicated.

embryonic kidney cells and mouse skeletal muscle this phenotype correlated with the observation that mitochondria were already swollen at steady state. We tested whether this mitochondrial swelling is also the case in *Drosophila Tmbim5* KO mitochondria by adding polyethylene glycol to a suspension of mitochondria prepared from whole adult flies. This treatment osmotically shrinks the mitochondria to their minimum size[33]. To compare the starting baseline, we normalized the absorbance to the value after PEG addition. This revealed that *Tmbim5* KO mitochondria are indeed pre-swollen (Fig. 2g), similar to mammalian TMBIM5 KO models. We next tested the total mitochondrial Ca²⁺ retention capacity by adding repetitive pulses of Ca²⁺ to isolated mitochondria. When the mitochondrial Ca²⁺ uptake capacity exceeds the organelle's buffering capacity, excessive Ca²⁺ accumulation can lead to the opening of the permeability transition pore (PTP), a non-selective high-conductance pore that dissipates mitochondrial membrane potential. This demonstrated a faster opening of the mPTP in *Tmbim5* KO mitochondria (Fig. 2h), suggesting an enhanced susceptibility of mutant mitochondria to this insult in line with the increased susceptibility to apoptosis and cytochrome *c* release in TMBIM5 loss-of-function cell models[18,19]. Based on these results, we consider the reduced total ATP levels to be caused by compromised mitochondrial function (disrupted cristae architecture, mitochondrial swelling, and enhanced mPTP susceptibility) that manifests right before the drop in lifespan. Together, these results imply that *Tmbim5* deficiency in flies results in the same pathophysiological changes

observed in mammalian systems. Tmbim5-deficient flies can thus serve as a bona fide model of TMBIM5 loss of function.

**Concomitant reduction of Micu1 mitigates lifespan reduction and attenuates mitochondrial defects in Tmbim5 knockout flies**

To investigate potential compensatory changes of the MCUC similar to what we observed in *TMBIM5*-deficient HEK293 cells, we then studied the expression levels of *Tmbim5* in flies deficient in *Mcu*, *Emre*, *Micu1*, and *Micu3*[28] and vice versa. This revealed an upregulation of *Mcu* mRNA in 4-7-day-old male *Tmbim5* KO flies but no changes in flies deficient in mRNA encoding the interacting proteins (Fig. S3a). This is in contrast to the slight downregulation of MCU protein in human *TMBIM5* KO cells (Fig. 1c), suggesting species-dependent differences or a post-translational regulation. We could not evaluate Tmbim5 protein levels in flies because it is not recognized by our antibodies. Interestingly, *Tmbim5* mRNA levels were also upregulated in *Mcu*-deficient flies, similar to *Mcu* being upregulated in *Tmbim5*-deficient flies; again, all other MCUC-deficient flies exhibited no statistically significant changes (Fig. S3b and c).

To define genetic interactions between the MCUC and Tmbim5, we next crossed *Tmbim5* KO males with MCUC-deficient flies. This drastically shortened the lifespan of double-deficient flies for *Mcu*, *Emre*, and *Micu3*-deficient flies (Fig. 3a). However, the reduction of *Micu1* levels in *Micu1 +/-* flies significantly increased the lifespan of wildtype and *Tmbim5* KO flies (Fig. 3a) and completely normalized the detrimental effect of *Tmbim5*

https://doi.org/10.1038/s42003-025-08839-6                                                                                                                  **Article**

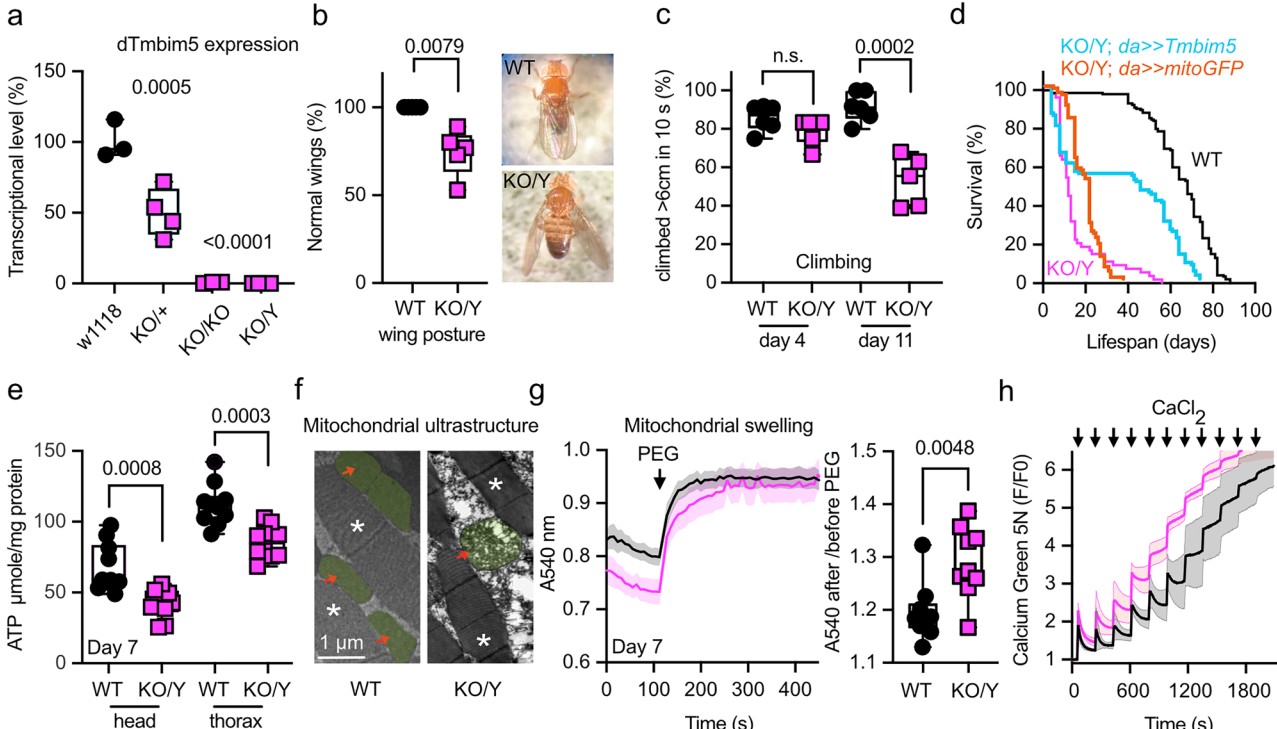

**Fig. 2 | Drosophila *Tmbim5* knockout impairs wing posture, mobility, and lifespan with phenotypes resembling mammalian models: altered cristae, mitochondrial swelling, and reduced Ca²⁺ buffering capacity. a** Tmbim5 expression assessed by quantitative PCR in wild type (WT, w[1118]), heterozygous (KO/+), homozygous (KO/KO), and hemizygous (KO/Y) knockout flies demonstrating loss of expression. Each data point contains five 4-7-day-old flies. **b** Abnormal wing posture in 6-week-old male KO/Y flies. Each data point contains the mean percentage of flies with normal wings in a sample of 25 flies. **c** Climbing ability of WT and KO/Y flies on the indicated days post-eclosion with an age-dependent decline in KO/Y. Each data point contains the mean speed of 25 flies. **d** Reduced lifespan of KO/Y flies, sample size >100 flies/group. Mild ubiquitous expression of Tmbim5 using *daughterless (da)*-gal4 but not mitoGFP in KO/Y flies confirmed that the lifespan defect was specifically due to Tmbim5 loss. **e** Ubiquitously reduced ATP levels in 7-day-old male *Tmbim5* knockout flies (KO/Y). Values were normalized to protein content. **f** Exemplary transmission electron microscopy images showing a disrupted cristae architecture of flight muscle mitochondria in 7-day-old KO/Y flies. Red arrows and green shading indicate mitochondria, asterisks mark indirect flight muscle. Size is indicated. **g** Larger delta in absorbance following osmotic shrinking with polyethylene glycol (PEG, added where indicated) in 7-day-old Tmbim5 KO mitochondria indicating swelling. Values were normalized to the last values after PEG-addition. Baseline absorbance and the ratio of absorbance before and after shrinking are quantified on the right. Data are shown as mean ± SEM, n = 9.
**h** Reduced mitochondrial calcium buffering capacity in 7-day-old KO/Y flies shown by quantifying Ca²⁺ uptake of isolated mitochondria in a bath with Calcium Green 5 N challenged by sequential addition of 5 μM CaCl₂ pulses. Fluorescence was normalized to the initial value (F₀). Data are shown as mean ± SEM, n = 5-9. Data are represented as box and whisker plots with the box representing the interquartile range, spanning from the 25th to the 75th percentile. A horizontal line within the box indicates the median. Whiskers extend from the minimum to the maximum data points. Each data point in A represented the mean value of 5 flies, in (**b, c**) 25 flies, in (**e**) 2 flies, in (**g** and **h**) 100 flies. Statistical significance was calculated by Šídák's multiple comparisons test in A, the Mann Whitney test in B, unpaired *t* tests in (**c, e, g**). *p*-values are indicated.

deficiency on climbing ability (Fig. 3b). This was also true when we used RNAi-mediated knockdown to reduce *Micu1* in *Tmbim5* KO flies (Fig. 3b). In line with the beneficial effect of *Micu1* depletion on lifespan and climbing ability of *Tmbim5*-deficient flies, *Micu1* heterozygosity also rescued the mitochondrial defects detected by transmission electron microscopy (Fig. 3c), attenuated the effects of *Tmbim5* KO on faster mPTP opening (Fig. 3d) and mitochondrial swelling (Fig. 3e). Together, these results imply that partial Micu1 reduction is largely beneficial and capable of mitigating the detrimental effects of Tmbim5 deficiency.

## TMBIM5 and MICU1 are found in the same macromolecular complex

To clarify whether the observed genetic interaction between MICU1 and TMBIM5 is mediated via physical proximity, we next analyzed the presence of TMBIM5 in macromolecular complexes containing MCU and MICU1 and assessed whether these complexes exhibit altered characteristics in the absence of TMBIM5. For this analysis, we utilized human cell models due to the availability of specific antibodies. We examined two human cell lines in which we previously generated and characterized *TMBIM5* knockout: HAP1[18] and HEK293[16]. Blue native gel electrophoresis revealed that in lysates from both cell lines, TMBIM5 and MICU1 predominantly migrated

in a complex of approximately 140 kDa, with minimal presence in a larger 480 kDa complex containing MCU. Notably, TMBIM5 deficiency did not alter the size or stability of MCU- or MICU1-containing complexes (Fig. 4a). These findings suggest the co-existence of MICU1 and TMBIM5 within a shared macromolecular complex. Consistent with this observation, we successfully co-immunoprecipitated native TMBIM5 with HA-tagged MICU1 but not with empty vector controls (Fig. 4b). Reciprocal co-immunoprecipitation experiments using native MICU1 and TMBIM5 were unsuccessful, likely due to antibody limitations. Collectively, these results indicate that MICU1 and TMBIM5 may exist in the same macromolecular complex and interact under certain conditions, providing a potential molecular basis for their observed genetic relationship.

## TMBIM5 and MICU1 deficiency results in similar changes in mitochondrial shape, size, and membrane potential compartmentalization

We next studied the effects of TMBIM5 and MICU1 deficiency on mitochondrial shape and size and found that both result in more rounded and smaller mitochondria with MICU1 knockout cells being even less tubular than *TMBIM5* knockout cells while maintaining a similar size (Fig. 4c). Besides its role as being the gatekeeper for the MCUC[4], MICU1 also

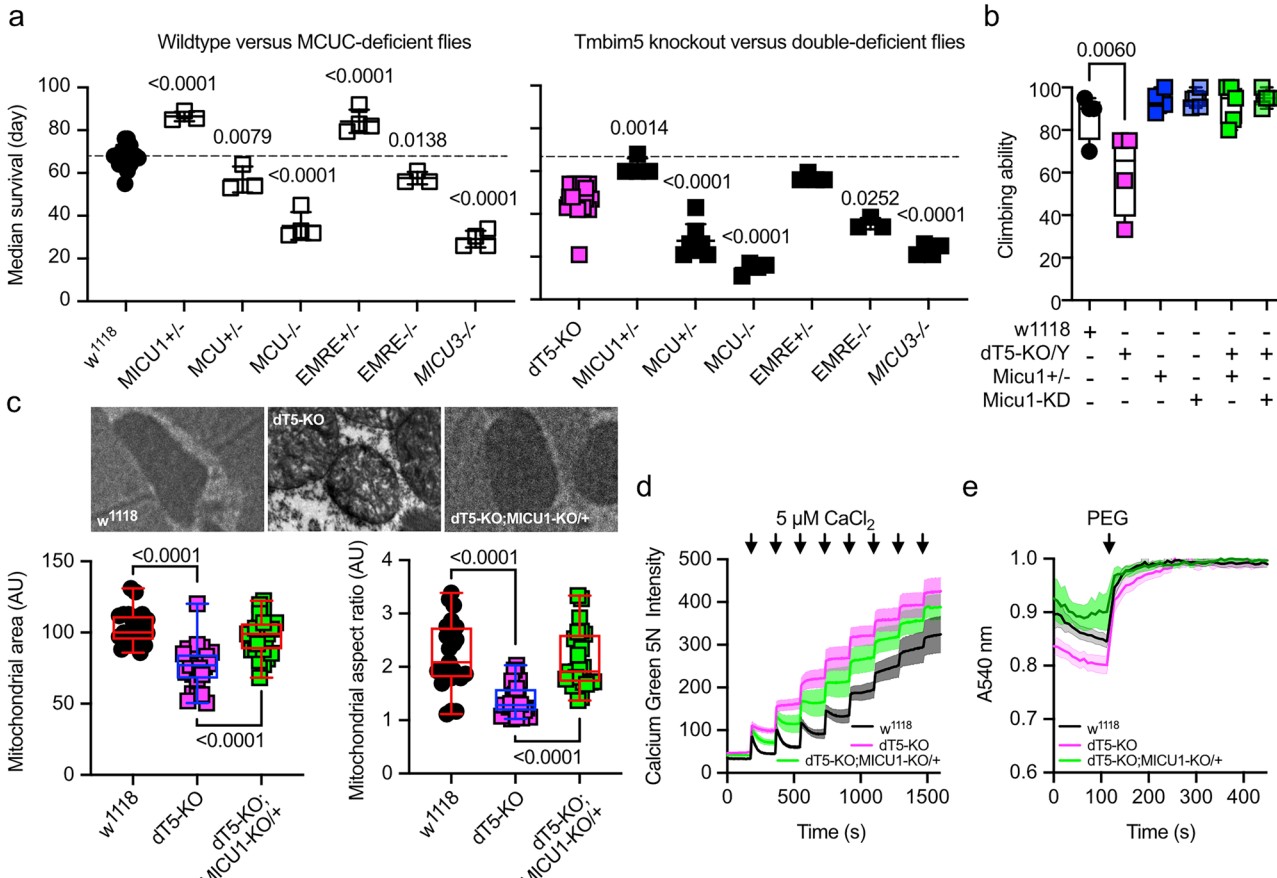

**Fig. 3 | Concomitant reduction of Micu1 mitigates lifespan reduction and attenuates mitochondrial defects in Tmbim5 knockout flies. a** Median survival of wild type ($w^{1118}$), *Tmbim5* knockout (*dT5-KO/Y*), MCUC-deficient flies compared with flies with a combined deficiency as indicated. DKO, double knockout. Double deficiency was detrimental for *Mcu*, *Emre*, *Micu3* and beneficial for *Micu1*. **b** Climbing ability of $w^{1118}$ (wild-type), *dTmbim5* knockout (*dT5-KO/Y*), with (+) or without (-) heterozygous *Micu1* knockout (*Micu1 + /-*), or ubiquitous *Micu1* knockdown using the GAL4-UAS system (*da-GAL4 > UAS-Micu1-RNAi*, abbreviated as *Micu1-KD*). **c** Mitochondrial area and aspect ratio of $w^{1118}$, *dT5-KO*, and *dT5-KO;Micu1 + /-* flies detected by transmission electron microscopy and analyzed by ImageJ. Typical pictures are shown. **d** Improved mitochondrial calcium buffering capacity in *dT5 KO/Micu1 + /-* double-deficient flies shown by quantifying $Ca^{2+}$ uptake of isolated mitochondria in a bath with Calcium Green 5 N challenged by sequential addition of 5 µM $CaCl_2$ pulses. Fluorescence was normalized to the initial value ($F_0$). Data are shown as mean ± SEM, *n* = 5-9. **e** Reduced swelling of *dT5 KO/ Micu1 + /-* double-deficient mitochondria shown by changes in absorbance following osmotic shrinking with polyethylene glycol (PEG, added where indicated). Values were normalized to the last values after PEG-addition. Baseline absorbance and the ratio of absorbance before and after shrinking are quantified on the right. Data are shown as mean ± SEM, *n* = 9. Each *n* represents a mitochondrial sample extracted from 100 flies. Data in (**a–c**) are presented as box and whisker plots with the box representing the interquartile range, spanning from the 25th to the 75th percentile. A horizontal line within the box indicates the median. Whiskers extend from the minimum to the maximum data points. Each data point in a and b represents the mean value of 25 flies. Statistical significance was determined using one-way ANOVA with Holm-Šídák's multiple comparisons test in (**a–c**), *p* values are indicated.

independently functions as a structural component of the inner mitochondrial membrane[24,25]. This membrane is structurally and functionally compartmentalized into two distinct domains: the cristae membrane (CM) and the inner boundary membrane (IBM). The IBM runs parallel to the outer mitochondrial membrane, maintaining a consistent distance and creating the intermembrane space, while the CM forms invaginations that project into the mitochondrial matrix. These domains are connected by narrow tubular structures called cristae junctions. MICU1 independently functions as a structural component of the IBM, where its exclusive localization through electrostatic interactions with cardiolipin stabilizes cristae junctions, maintains mitochondrial membrane potential, and prevents cytochrome *c* redistribution from cristae to the intermembrane space[24]. TMBIM5 is also essential for maintaining cristae architecture[16,18,19] and preventing cytochrome c release[19]. Both proteins also interact with CHCHD2[20,25], a component of the MICOS. We, therefore, figured that they most probably functionally interact in their role in the maintenance of the IBM. We, therefore, quantified the IBM association index plotted against descending concentrations of TMRM in wildtype, TMBIM5 KO, and

MICU1 KO cells to determine the relative membrane potential distribution between IBM and CM. This was achieved by using structured illumination microscopy, which allows the visualization and precise localization of the membrane potential indicator dye TMRM within sub-mitochondrial compartments, and thus a visualization between the highly stained, highly polarized CM and lower stained, lower polarized IBM[34]. A higher IBM association index indicates relatively more TMRM accumulation in the IBM compared to the highly stained CM, reflecting a homogenization in membrane potential between these two compartments. Using this analysis, we observed a reduced TMRM concentration for half-maximal changes in the IBM association index in TMBIM5 and MICU1 KO cells (Fig. 4d). These changes were more pronounced in MICU1-deficient cells (Fig. 4d). These data indicate that the knockdown of either TMBIM5 or MICU1 results in an equilibration of the membrane potential between cristae and IBM. A deficiency of both proteins resulted in a left shift of the IBM association index, indicating compromised electrochemical compartmentalization between mitochondrial subdomains (Fig. 4d). This phenomenon reflects impaired maintenance of the membrane potential gradient that normally exists

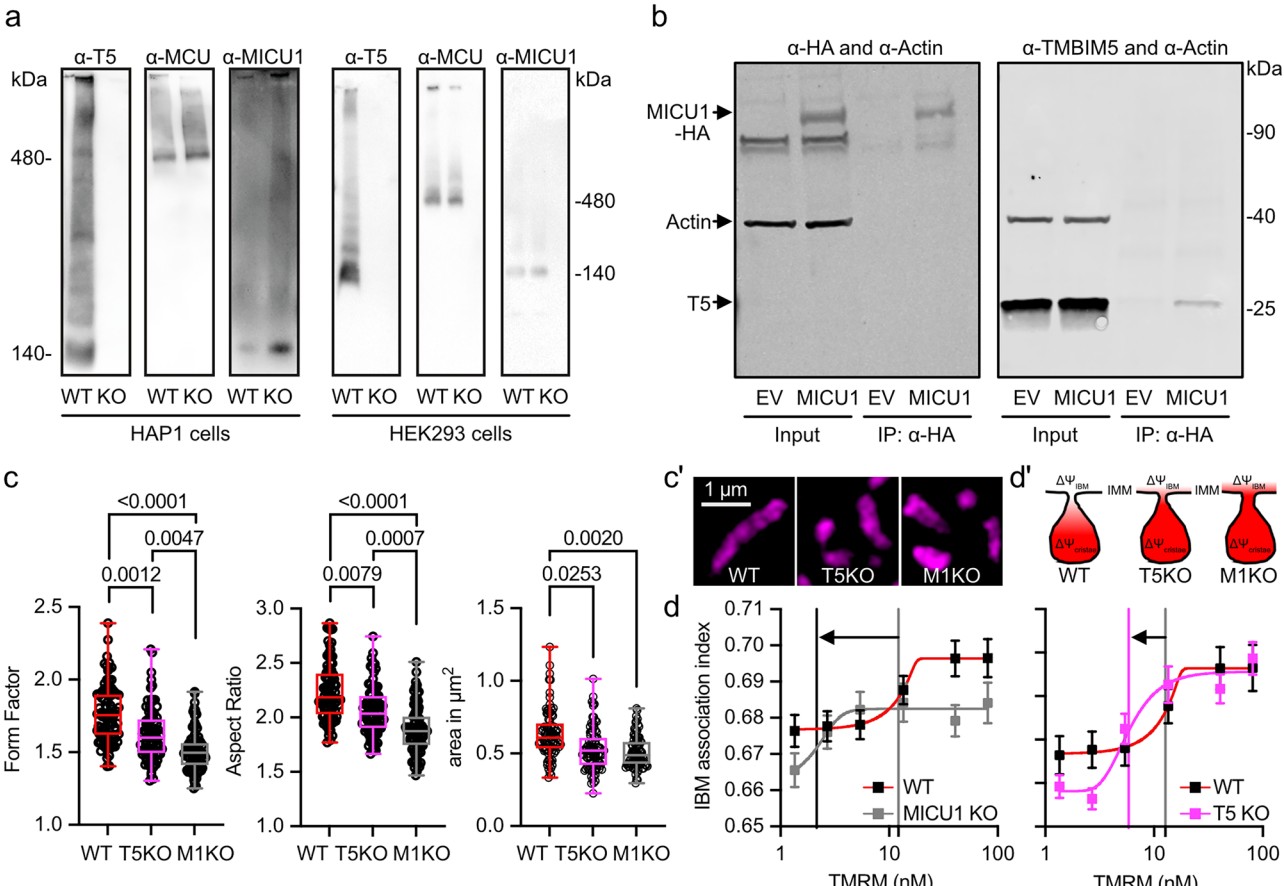

**Fig. 4 | TMBIM5 and MICU1 are found in the same macromolecular complex and their deficiency results in similar changes in mitochondrial shape and inner boundary membrane. a** Blue native gel electrophoresis of TMBIM5, MCU and MICU1 in human HAP1 (left) and HEK293 cells with (wild type, WT) and without TMBIM5 (knockout, KO) expression. Approximate molecular weight is indicated. **b** Co-immunoprecipitation of TMBIM5 with HA-tagged MICU1 (arrow). Left panel shows successful pull-down of tagged MICU1 and the right panel coimmunoprecipitation of endogenous TMBIM5. Molecular weight is indicated. **c/c'** Morphometric analysis of TMRM-labeled mitochondria in WT, TMBIM5 (T5) KO, and MICU1 (M1) KO HEK cells imaged with structured illumination microscopy. **C'** shows typical pictures, size is indicated. **d/d'** Evaluation of mitochondrial membrane potential (ΔΨm) gradients of WT, T5KO, and M1KO HEK cells using structured illumination microscopy. **d'** illustrates the findings. Vertical lines in D indicate the TMRM concentration threshold for half-maximal IBM association index changes. Lower thresholds reflect more even membrane potential distribution between cristae and inner boundary membrane; higher thresholds indicate greater potential differences between these compartments. Data in (**c**) are presented as box and whisker plots with the box representing the interquartile range, spanning from the 25th to the 75th percentile. Whiskers extend from the minimum to the maximum data points. Each data point represents a single cell accumulated from 4 independent experiments. Data in (**d**) show the mean ± standard errors of mean. Statistical significance in c was determined using the Kruskal-Wallis test followed by Dunn's multiple comparisons test, *p* values are indicated.

between the IBM and CM. Together, this implies that TMBIM5 and MICU1 functionally converge in maintaining mitochondrial shape, size, and membrane potential compartmentalization.

## MICU1 and TMBIM5 exhibit reciprocal effects on mitochondrial morphology and submitochondrial localization

We next investigated whether TMBIM5 can mitigate changes observed in MICU1 knockout and vice versa to establish whether the two proteins work cooperatively and to clarify which protein is upstream and which is downstream. We found that overexpression of MICU1-GFP rescued the rounder shape quantified by form factor and aspect ratio (Fig. 5a) of *TMBIM5* KO cells. Overexpression of TMBIM5-GFP or a channel pore mutant D294R/D325R (DM, described in[16]), in contrast, had no effect on the roundish shape of mitochondria in *MICU1* KO cells (Fig. 5b). A similar pattern emerged when we studied mitochondrial size. MICU1-GFP again rescued the smaller size of *TMBIM5* KO mitochondria (Fig. 5c), while both TMBIM5-GFP and DM-TMBIM5-GFP surprisingly even further reduced the smaller size of MICU1 KO cells (Fig. 5d). In wildtype mitochondria, only DM-TMBIM5 reduced the size of mitochondria, while TMBIM5 had no effect (Fig. 5d).

When examining submitochondrial localization using the IBM association index methodology[34], we observed patterns that contrasted with the morphological effects. Overexpression of MICU1-GFP in wildtype cells resulted in a highly positive association index, indicating predominant localization at the inner boundary membrane as described[24]. However, this spatial organization was significantly disrupted in TMBIM5 KO cells (Fig. 5e), despite MICU1-GFP's ability to rescue mitochondrial size in this background. Conversely, TMBIM5-GFP and DM-TMBIM5-GFP showed no positive correlation with the IBM in wildtype cells, but both variants shifted to the inner boundary membrane in MICU1 KO cells (Fig. 5f), where they exacerbated size reduction.

These opposing patterns reveal an inverse correlation between mitochondrial membrane compartmentalization and morphological regulation. In TMBIM5 KO cells, MICU1 overexpression results in more branched mitochondria despite lower association with the IBM. In MICU1 KO cells, TMBIM5 results in smaller mitochondria while showing a higher association with the IBM, irrespective of its channel function. Together, these findings suggest that TMBIM5 might function upstream of MICU1 in the regulation of mitochondrial morphology, as MICU1 can rescue phenotypes caused by TMBIM5 deficiency. The channel pore mutant DM-TMBIM5

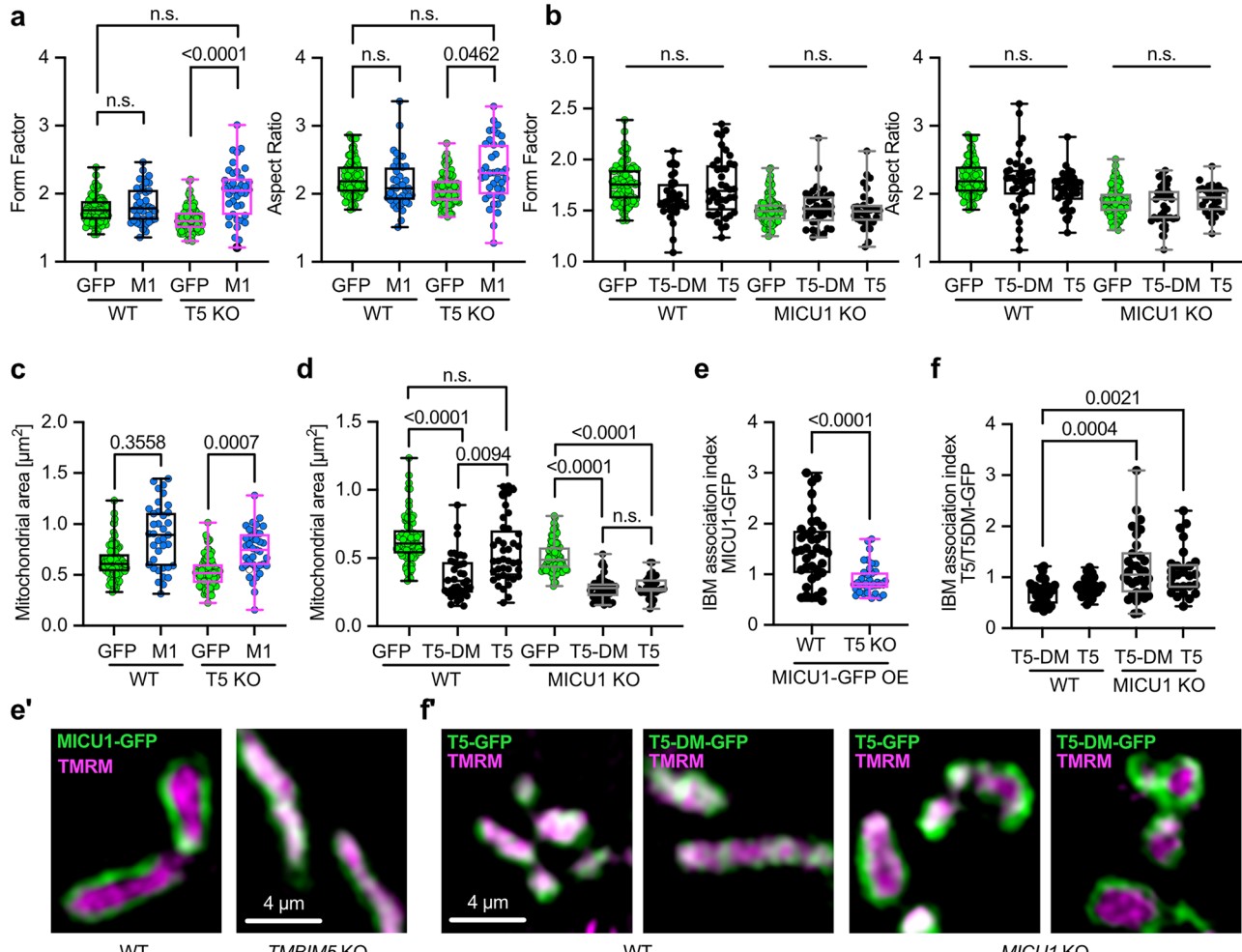

**Fig. 5 | MICU1 and TMBIM5 exhibit reciprocal effects on mitochondrial morphology and submitochondrial localization. a** Morphometric analysis of TMRM labeled mitochondria in WT and T5KO HEK cells expressing control vector (GFP) or MICU1-GFP (M1) imaged with structured illumination microscopy **b** Morphometric analysis of TMRM labeled mitochondria in WT and M1KO HEK cells expressing control vector (GFP), TMBIM5-GFP (T5), or TMBIM5DM-GFP (T5DM) imaged with structured illumination microscopy. **c** Analysis of mitochondrial area using TMRM labelling in WT and T5KO HEK cells expressing control vector (GFP) or MICU1-GFP (M1) imaged with structured illumination microscopy. **d** Analysis of mitochondrial area using TMRM labelling in WT and M1KO HEK cells expressing control vector (GFP), TMBIM5-GFP (T5), or TMBIM5DM-GFP (T5DM) imaged with structured illumination microscopy. **e/e'** Analysis of the submitochondrial localization of MICU1-GFP using the IBM association factor with TMRM as reference label in WT and T5KO HEK cells.

Representative images of TMRM stained WT and T5KO HEK cells expressing MICU1-GFP (M1) imaged with structured illumination microscopy. **f/f'** Analysis of the submitochondrial localization of TMBIM5-GFP and TMBIM5DM-GFP using the IBM association factor with TMRM as reference label in WT and M1KO HEK cells. Representative images of TMRM stained WT and M1KO HEK cells expressing TMBIM5-GFP (T5-GFP) or TMBIM5DM-GFP (T5DM-GFP) imaged with structured illumination microscopy. Data are presented as box and whisker plots with the box representing the interquartile range, spanning from the 25th to the 75th percentile. A horizontal line within the box indicates the median. Whiskers extend from the minimum to the maximum data points. Each data point represents a single cell accumulated from 4 independent experiments. Statistical significance in (**a–d** and **f**) was determined using the Kruskal-Wallis test followed by Dunn's multiple comparisons test and in (**e**) using unpaired *t* test, *p* values are indicated.

exhibits dominant-negative effects by reducing mitochondrial size in wild-type cells, whereas wild-type TMBIM5 does not. However, in MICU1 KO cells, both wild-type and mutant TMBIM5 reduce mitochondrial size similarly, indicating that MICU1 may normally suppress a channel-independent, size-reducing function of TMBIM5 that becomes dysregulated when MICU1 is absent.

## Discussion

In this study, we characterized the interaction between TMBIM5—a bidirectional pH-sensitive Ca²⁺ transport system in the inner mitochondrial membrane—and components of the mitochondrial calcium uniporter complex. Genetic studies in *Drosophila melanogaster* revealed that partial MICU1 depletion selectively rescued phenotypes in Tmbim5-deficient flies, which otherwise display disrupted mitochondrial structure and function similar to findings observed in mice. In mammalian cells, TMBIM5 and

MICU1 were found in the same macromolecular complexes, and their deficiency results in comparable alterations in mitochondrial morphology and membrane potential compartmentalization. Complementation assays demonstrated that MICU1 expression restored mitochondrial morphology in TMBIM5-knockout cells, while TMBIM5 overexpression exacerbated mitochondrial defects in MICU1-deficient cells, with both proteins exhibiting opposing effects on submitochondrial distribution. Together, this suggests a functional interplay between TMBIM5 and MICU1 that maintains mitochondrial structural and functional homeostasis.

We envisage two potential hypotheses how partial MICU1 depletion could ameliorate TMBIM5-deficiency phenotypes. Either MICU1 is directly involved in the mechanism that leads to TMBIM5-dependent mitochondrial dysfunction, or partial MICU1 depletion triggers a cellular stress response program that also protects against TMBIM5 deficiency. A direct involvement could theoretically involve MICU1's effect on mitochondrial

$Ca^{2+}$ entry. MICU1 deficiency results in increased mitochondrial $Ca^{2+}$ levels because loss of its inhibitory "gatekeeping" function[4] allows constitutive $Ca^{2+}$ leak into the mitochondria even at resting/low cytosolic $Ca^{2+}$ concentrations, resulting in increased mitochondrial $Ca^{2+}$ levels. It is unlikely that a combined MICU1 and TMBIM5 deficiency would ameliorate these potentially detrimental $Ca^{2+}$ levels as TMBIM5 itself is also involved in mitochondrial efflux[14,15]. Also, double deficiency of TMBIM5 with MCU and EMRE, which would lower mitochondrial $Ca^{2+}$ levels, was detrimental.

So other, non-MCUC functions of MICU1 could be more relevant. MICU1 functions as a critical regulator in cold-induced ferroptosis[35], a form of regulated cell death mediated by iron-dependent lipid peroxidation[36]. This role is distinct from its canonical $Ca^{2+}$ gatekeeping role and involves the prevention of mitochondrial hyperpolarization. A structure-function analysis revealed that domains involved in MICU1 dimerization, rather than the $Ca^{2+}$-sensing EF-hand domains, were essential for this process, indicating a primarily structural contribution to ferroptosis regulation. In our TMBIM5-deficient model, we observed a constellation of mitochondrial abnormalities—disrupted cristae architecture, organelle swelling, and enhanced susceptibility to permeability transition—that collectively create cellular conditions conducive to ferroptotic processes. It is, therefore, reasonable that partial MICU1 reduction could attenuate these pathological effects by specifically preventing the mitochondrial hyperpolarization established as a prerequisite for lipid ROS accumulation during ferroptosis, thereby providing a mechanistic explanation for the rescue phenotype observed in our double-deficient model. Consistent with our findings, only MICU1 but not MICU2 seems to be important in this pathway[35]. Furthermore, an involvement of TMBIM5 in hyperpolarization-triggered ferroptosis mediated by MICU1 is in line with the findings described by Patron et al. that persistent mitochondrial hyperpolarization initiates a regulatory cascade beginning with TMBIM5 degradation[15]. This degradation subsequently releases the mitochondrial m-AAA protease AFG3L2 from inhibition, enabling its proteolytic activity. The unleashed AFG3L2 then orchestrates a comprehensive remodeling of the mitochondrial proteome, particularly targeting respiratory complex I subunits. This proteolytic breakdown of complex I components serves as a compensatory mechanism to attenuate excessive hyperpolarization by limiting electron transport chain activity and reducing reactive oxygen species production - essentially creating a negative feedback loop to restore membrane potential homeostasis under conditions of persistent hyperpolarization[15].

Alternatively, the hypothesis that MICU1 triggers a protective stress response, which also protects against TMBIM5 deficiency, appears plausible. Such a stress response could involve activation of the DELE1 (DAP3-binding cell death enhancer 1) pathway through the altered cristae structure observed in MICU1-depleted flies. In this scenario, changes in cristae architecture would activate the protease OMA1, leading to proteolytic processing of DELE1 and translocation to the cytosol, culminating in activation of the mitochondrial integrated stress response[37,38]. Interestingly, this protection is particularly evident in skeletal muscle, which parallels the tissue-specific vulnerability observed in both TMBIM5 and MICU1 deficiency models[39].

The paradoxical observation that TMBIM5 and MICU1 exhibit opposing effects on submitochondrial localization and mitochondrial morphology reveals complex regulatory dynamics within the inner mitochondrial membrane. We observed contrasting effects on homogenization of mitochondrial membrane potential and morphological rescue effects, suggesting a dissociation between localization and functional outcomes. Both TMBIM5 and MICU1 interact with CHCHD2 and MIC60[20,25], components of the MICOS complex independently of the MCUC, which explains their presence in shared macromolecular complexes despite distinct primary functions. When one protein is absent, the other one might undergo compensatory redistribution – TMBIM5 shifts to the inner boundary membrane in MICU1-knockout cells, while MICU1's localization depends on TMBIM5 presence. This functional overlap in maintaining cristae architecture and preventing cytochrome *c* release, combined with their context-dependent redistribution, suggests that proper mitochondrial

function requires coordinated spatial organization of both proteins within the inner mitochondrial membrane. However, a clear hierarchy cannot be established because the observed rescue effects could result from functional redundancy or compensatory mechanisms rather than true upstream/downstream regulation, and the reciprocal localization dependencies suggest mutual interdependence rather than unidirectional control.

A particularly intriguing mechanistic hypothesis that should be addressed in future work involves MICU1 functioning as an intermediary signal transducer between the IMS $Ca^{2+}$ environment and TMBIM5 activity. Given that MICU1 contains $Ca^{2+}$-sensing EF-hand domains and our demonstration that MICU1 and TMBIM5 exist within the same macromolecular complex, MICU1 could potentially confer information about IMS $Ca^{2+}$ concentrations directly to TMBIM5. In this model, conformational changes in MICU1 induced by $Ca^{2+}$ binding would modulate TMBIM5's bidirectional $Ca^{2+}/H^+$ exchange activity through direct protein-protein interactions. The reciprocal effects observed in our complementation studies, where MICU1 expression rescues TMBIM5-knockout phenotypes while TMBIM5 exacerbates MICU1-deficient mitochondrial abnormalities, support such a hierarchical regulatory relationship. This proposed mechanism aligns with previous observations that MICU1 has functional roles beyond MCU regulation in *Drosophila*, as complete MICU1 deficiency caused lethality that could not be rescued by simultaneous MCU knockout[28]. Our findings extend this concept by suggesting that these MCU-independent functions of MICU1 may include direct modulation of TMBIM5-mediated $Ca^{2+}$ transport. This would establish a sophisticated regulatory circuit for fine-tuning mitochondrial $Ca^{2+}$ homeostasis independent of the canonical MCU-mediated uptake pathway.

In conclusion, our findings establish a functional interplay between TMBIM5 and MICU1 in maintaining mitochondrial integrity, with implications for understanding the molecular basis of mitochondrial disorders, particularly those affecting skeletal muscle.

## Materials and methods
### Denaturing immunoblotting

To obtain protein samples, cells were directly lysed in Dodecyl-β-D-maltoside-lysis buffer (DDM-lysis buffer; 50 mM HEPES, 150 mM NaCl, 0.2% DDM, 0.5 mM EGTA, 0.3 mM $CaCl_2$). Mouse tissue samples were homogenized in the same buffer at 4000 x rpm, 30 s using a glass-Teflon-potter (1-3x, until homogeneous). After 30 minutes (min) solubilisation (4 °C, rotating), all samples were centrifuged (21,000 x *g*, 10 min, 4°C) and the supernatant was used for western blotting. Protein samples were denatured in 1x Laemmli-β-mercaptoethanol-buffer, 95 °C, 5 min. After gel electrophoretic separation of the proteins, they were transferred to nitrocellulose membranes by using a semi-dry blotting system (Bio-rad). For the quantification of TMBIM5, membranes were incubated in SDS-β-mercaptoethanol solution (100 mM β-mercaptoethanol, 2% SDS, 62.5 mM Tris-HCl, pH 6.7) at 55 °C for 15 min on a shaker. After washing (TBST) and blocking (3% milk powder in TBST, 1 hour (h), room temperature (RT)), the membranes were incubated with the respective primary antibodies (overnight, 4 °C, rotating). Antibodies used: rabbit anti-TMBIM5 (Proteintech, 1:1,000), rabbit anti-MCU (Millipore Sigma, 1:500), rabbit polyclonal anti-MICU1 (Merck, 1:500), rabbit polyclonal anti-MICU2 (Abcam, 1:500), rabbit polyclonal anti-alpha-LETM1 (Thermo Fisher, 1:500), rabbit polyclonal anti-EMRE (Santa Cruz, 1:200), rabbit anti-HA (Abcam, 1:500), mouse anti-actin (Merck chemicals, 1:1,000). Fluorescence-labelled secondary antibodies were used, and the signal was detected using a Li-Cor Odyssey imaging system and quantified with the Image Studio Lite software. The intensity was normalized to the loading control and the mean per blot.

### Blue native polyacrylamide gel electrophoresis (BN PAGE)
Protein samples were solubilised with 5% digitonin on ice for 15 min followed by centrifugation (20,000 x *g*, 30 min, 4 °C). 0.25% of G-250 was added to the supernatant, and complexes were separated via 4- 16% Bis-Tris gels (NativePAGE ™, Thermo Fisher). The complexes were transferred to

PVDF-membranes via wet blotting without methanol. Following fixation (8% acetic acid), destaining (50% methanol and 25% acetic acid), and blocking (3% milk powder in TBST, 1 h, RT), the membranes were incubated with the respective primary antibodies (overnight, 4 °C, rotating). The used antibodies are listed above. The staining of HRP-coupled secondary antibodies was detected with the Clarity ™ Western ECL Kit (Bio-Rad)/ SuperSignal™ West Femto Maximum sensitivity Kit (Thermo Fisher Scientific) using a Bio-Rad detection system.

### Fly stocks
*Tmbim5/CG2076* KO flies were obtained from Wellgenetics, Taiwan. *MCU, EMRE, MICU1,* and *MICU3* KO flies were provided by Alex Whitworth of the MRC Mitochondrial Biology Unit, Cambridge, UK. The UAS-MICU1-RNAi line was obtained from Bloomington #419069. All flies were maintained at 25 °C on standard food.

### Lifespan
Fruit flies were provided with standard molasses-based food and accommodated in a climate chamber at a temperature of 25 °C with a 12 h light and 12 h darkness schedule. The flies were given fresh food every two days and dead flies were scored. Each experiment consisted of at least four groups of 25 flies in each group.

### Climbing
Climbing assays were conducted on at least four groups of 25 flies per genotype in 2 cm diameter vials, recording the percentage of flies climbing 6 cm in 10 s. Tests were standardized for time of day without $CO_2$ anesthesia 24 h prior.

### Wing phenotype
Abnormal wings were scored in six-week-old flies across five groups per genotype, with each group containing 25 flies.

### ATP content
ATP levels in flies were measured following a method described previously[40]. Briefly, two heads, thoraxes, or abdomens were lysed in 100 µl of lysis buffer from the ATP Bioluminescence Assay Kit HS II (Merck 11699709001), heated at 95 °C for 2 min, and centrifuged at maximum speed at 4 °C for 1 min. The assay mixed 2.5 µl of the clear supernatant, 187.5 µl of dilution buffer, and 10 µl of luciferase from the kit, and luminescence was immediately measured using a Spark Multimode Microplate Reader (Tecan). ATP concentrations were quantified using an ATP standard curve and normalized to protein concentrations measured via the BC Assay.

### Quantification of mitochondrial and nuclear DNA
Total DNA, including mtDNA, was purified using the Quick-gDNA Miniprep Kit (ZYMO RESEARCH, D3025), designed for isolating various DNA types such as genomic and mitochondrial. The mtDNA copy number was estimated by comparing the mtCOI mitochondrial gene to the nuclear gene nCOX5A, both encoding proteins involved in the electron transport chain. Detection was performed using the Faststart Universal SYBR Green Master kit (Sigma-Aldrich 4913850001) with each reaction containing 10 ng of DNA and 10 pmol of each primer. mtDNA levels were quantified using the $2^{-\Delta\Delta Ct}$ method, where $\Delta Ct = Ct\_mtCOI - Ct\_nCOX5A$ and $\Delta\Delta Ct = \Delta Ct$ experimental group - mean $\Delta Ct$ control group.

### Functional mitochondrial preparation
Functional mitochondrial preparation followed published protocols[28,41]. Briefly, 100 flies were homogenized in 2 ml of Mannitol-Sucrose (MS) buffer (containing 225 mM mannitol, 75 mM sucrose, 5 mM HEPES, and 0.1 mM EGTA/Tris, pH adjusted to 7.4 with 10 M KOH, all stored at 4 °C) supplemented with 1% bovine serum albumin (BSA). The homogenization was conducted using a Dounce glass potter with a loose-fitting glass pestle at a speed setting of 1.5 for 15 strokes. The homogenate was centrifuged at 1000 x g for 10 min at 4 °C, and the supernatant was filtered through a fine

mesh (100 µm). This filtrate was then centrifuged twice at 6000 x g for 10 min each at 4 °C in 5 ml of MS buffer containing BSA. The resultant pellet was resuspended in 1.5 ml of incubation buffer (10 mM Tris/MOPs, 10 µM EGTA/Tris, 5 mM Pi/Tris, 5 mM glutamate, 2.5 mM malate, and 250 mM sucrose, pH adjusted to 7.4 with HCl, all stored at 4 °C) and centrifuged at 7000 x g for 10 min at 4 °C. The final pellet was resuspended in 60 µl of the same incubation buffer. Protein concentration was then assessed using a BCA assay with a 1:100 dilution.

### Citrate synthase activity
Citrate synthase activity was assessed in whole protein lysates. The assay was performed in 96 well plates as follows: 2.5 µl of Triton 10%, plus 2.5 µl of acetyl CoA 12.2 mM, plus 10 µl of DTNB (5,5´-dithiobis-(2-nitrobenzoic acid) 1 mM were added to each well. Then, 2 µl of protein lysate was added together with 78 µl of dH2O. The plate was placed in a TECAN Infinite Pro 200 reader, and the baseline acquisition was started. The absorbance was measured at OD 412 nm. After baseline acquisition, 5 µl of oxaloacetate (10 mM, pH 8.0) was added quickly to each well, and the instrument immediately started the absorbance measurements in kinetic mode at RT for 3 min in 10 s intervals.

### Transmission electron microscopy
Fly thoraxes were fixed using a solution containing 0.2 M cacodylate buffer (Caco-buffer), 25% glutaraldehyde (GA), and 15% paraformaldehyde (PFA). Post-fixation, tissues were washed in a 0.1 M Caco-buffer solution diluted with ddH2O and then incubated in 2% osmium tetroxide with 0.2 M buffer. After a second wash, samples were dehydrated in an ethanol gradient (30–70%) and stored overnight at 4°C. Dehydration continued the following day using 80%, 90%, 95%, and 100% ethanol, followed by washing with propylene oxide and overnight incubation in a 1:1 mixture of resin and propylene oxide. The next day, tissues were transferred to pure resin, embedded in molds, and polymerized in an oven for two days. The prepared samples were sectioned, stained for contrast, and imaged using a Tecnai 12 Transmission Electron Microscope.

### Mitochondrial PEG shrinking assay
Functional mitochondria were resuspended in assay buffer (125 mM KCl, 10 mM HEPES, 2 mM MgCl2, 2 mM K2HPO4, pH 7.2 adjusted with KOH), freshly supplemented with 100 mM succinate and 0.2 µM thapsigargin. Following a 2 min baseline measurement, polyethylene glycol-3350 (PEG, 5% final concentration) was added to induce shrinkage in pre-swollen mitochondria, requiring a large volume (100 µl) due to PEG's high viscosity. Measurements were normalized to the final min post-PEG addition to account for baseline variations. Absorbance was recorded at 540 nm using a Spark Multimode Microplate Reader.

### Quantitative RT-PCR
Total RNA from 4 to 7-day-old male flies was extracted using the ZR RNA MiniPrep kit (ZYMO RESEARCH), and cDNA was synthesized from 10 ng/µl RNA using the High Capacity cDNA Reverse Transcription Kits (Life Tech). Quantitative PCR (qPCR) was performed with FastStart Universal SYBR Green Master (Rox) (Merck), employing primers sourced from Eurofins. Transcriptional levels were quantified using the $2^{-\Delta\Delta Ct}$ method, where $\Delta\Delta Ct = \Delta Ct$ of the experimental group minus the mean $\Delta Ct$ of control groups. $\Delta Ct$ was calculated as Ct of the gene of interest minus Ct of the housekeeping genes RpL32/Rp49, which served as normalization controls.

### Cell culture
Cells were washed once with loading-buffer containing in mM: 2 CaCl2, 135 NaCl, 5 KCl, 1 MgCl2, 1 HEPES, 2.6 NaHCO3, 0.44 KH2PO4, 0.34 Na2HPO4, 10 D-glucose (Carl Roth, Karlsruhe, Germany), 0.1% vitamins, 0.2% essential amino acids and 1% penicillin/streptomycin at pH 7.4. Cells were incubated in loading-buffer containing 81, 40.5, 13.5, 5.4, 2.7 or 1.35 nM TMRM (tetramethylrhodamine methyl ester, Invitrogen™) and

500 nM MitoTracker™ Green FM for 60 min. As TMRM might degrade over time in storage, TMRM concentrations were measured regularly as described elsewhere[34].

## Structured illumination microscopy

Single and dual camera SIM imaging. The SIM setup used is composed of a 405 nm, 488 nm, 515 nm, 532 nm, and a 561 nm excitation laser introduced at the back focal plane inside the SIM-box with a multimodal optical fiber. For super-resolution, a CFI SR Apochromat TIRF 100x-oil (NA 1.49) objective was mounted on a Nikon-Structured Illumination Microscopy (N-SIM®, Nikon, Austria) System with standard wide field and SIM filter-sets and equipped with two Andor iXon3® EMCCD cameras mounted to a Two Camera Imaging Adapter (NikonAustria, Vienna, Austria). At the bottom port a third CCD-camera (CoolSNAP HQ2, Photometrics, Tucson, USA) is mounted for wide-field imaging. For calibration and reconstruction of SIM images the Nikon software (NIS-Elements AR 4.51.00 64-bit, Nikon, Austria) was used. Reconstruction was permanently performed with the same robust setting to avoid artefact generation and ensured reproducibility with a small loss of resolution of 10% compared to most sensitive and rigorous reconstruction settings. Microscopy setup adjustments were done as described elsewhere[24].

## IBM association index

The IBM association factor of TMRM was calculated as described elsewhere[34]. In short, images were subjected to background subtraction (Mosaic Suite, background subtractor, NIH) with a sliding rectangle diameter of 50 pixel. The reference channel (MTG, TMRM) was Otsu[42] auto thresholded and further dilated and eroded in two independent subsets. 1 erosion and 2 dilation iterations were used. Pixel-wise subtraction of the erosion reference of the dilated reference image yields in a hollow structure, used as a mask to measure the mean intensity in the mitochondrial periphery or IBM related area in the object channel. The erosion reference served as a mask to measure the bulk or cristae mean fluorescence intensity. The ratio of IBM/CM mean intensity is a value to estimate changes of the object label distribution inside a mitochondrion, which is referred to as the IBM association index. The higher the ratio value the higher the distribution of protein label in the IBM. For image analysis, the freeware program ImageJ was used[43].

## Mitochondrial morphology

Single 3D-SIM and time-lapsed images of TMRM were used for morphological analysis. Images were background corrected with an ImageJ Plugin (Mosaic Suite, background subtractor, NIH) and a binarization was done using a Yen auto threshold[44]. The ImageJ particle analyzer was used to extract the mitochondrial count (c), area (a), perimeter (p), minor (x) and major (y) axes of the mitochondria. Aspect ratio (AR) was determined as:

$$AR = \frac{y}{x}$$

The form factor (FF) was determined as follows:

$$FF = \frac{p^2}{4\pi \cdot a}$$

## Statistics and reproducibility

Graphpad Prism was used for statistical analyses. Outliers were removed using the ROUT algorithm and normality determined using the D'Agostino-Pearson test. This was followed by specific tests as indicated to determine significant differences depending on the number of conditions to be compared e.g. two conditions, parametrically distributed student's $t$ test, non-parametrically distributed Mann Whitney test, more than two conditions parametrically distributed ANOVA, non-parametrically distributed Kruskal Wallis test. A difference with a $p < 0.05$ was considered as statistically significantly different.

| Reagent or resource | Source | Identifier |
|---|---|---|
| **Antibodies** | | |
| Rabbit polyclonal anti-TMBIM5 (GHITM) | Proteintech | Cat#16296-1-AP; RRID:AB_2111275 |
| Rabbit polyclonal anti-MCU | Millipore Sigma | Cat#HPA016480; RRID:AB_2071893 |
| Rabbit polyclonal anti-MICU1 | Merck | Cat#HPA037479; RRID:AB_2675495 |
| Rabbit polyclonal anti-MICU2 | Abcam | Cat#ab101465; RRID: AB_10711219 |
| Rabbit polyclonal anti-alpha-LETM1 | Thermo Fisher | Cat#PA5-22233; RRID:AB_11152727 |
| Rabbit polyclonal anti-EMRE | Santa Cruz | Cat#sc-86337; RRID:AB_2250685 |
| Rabbit anti-HA | Abcam | Cat#ab9110-100; RRID:AB_307019 |
| Mouse monoclonal anti-alpha-actin | Merck chemicals | Cat#MAB1501; RRID:AB_2223041 |
| Anti-mouse IgG (H + L) (DyLight 680 Conjugate) | Cell Signaling | Cat#5470S; RRID:AB_10696895 |
| Anti-rabbit IgG (H + L) (DyLight 800 Conjugate) | Cell Signaling | Cat#5151S; RRID:AB_10697505 |
| **Bacterial and virus strains** | | |
| Escherichia coli DH5α competent cells | own lab | |
| **Chemicals, peptides, and recombinant proteins** | | |
| HEPES | Merck | Cat#H3375-100G |
| NaCl | Roth | Cat#3957.1 |
| Dodecyl-beta-D-maltoside (DDM) | Roth | CAS Nr. 69227-93-6 |
| Laemmli buffer | Bio-Rad | Cat##1610747 |
| BN PAGE loading dye | Invitrogen/ Thermo Fisher | |
| EGTA | Roth | Cat#30542 |
| $CaCl_2$ | Roth | Cat#C3306-500G |
| β-Mercaptoethanol | Bio-Rad | |
| SDS | Sigma Aldrich | Cat#L3771-25G (51719236) |
| Tris | Roth | Cat#4855.2 |
| HCl | Roth | Cat#T134.1 (51742804) |
| milk powder | Roth | Cat#T145.1 |
| Bovine Serum Albumin | Sigma-Aldrich | Cat#A7906-100G (51153600) |
| Sucrose | Sigma-Aldrich | Cat#84100-250 G |
| D-Mannitol | Sigma-Aldrich | Cat#M4125-500g |
| MOPS | Sigma-Aldrich | Cat#M1254-25G |

| | | | | | |
|---|---|---|---|---|---|
| L-Glutamic acid | Sigma-Aldrich | Cat#G1251-100G | w;EMRE[1]/(Cyo.GFP) (EMRE knockout fly) | 28 | |
| Malate | Sigma-Aldrich | M107-5MG | w[1118],dTmbim5-KO;;da-GAL4 | this paper | |
| Monopotassium phosphate | Roth | Cat#3904.1 (51465563) | w[1118];;da-GAL4 | Gift of Olaf Vef | |
| Fetal bovine serum | HyClone | Cat#1235-0273 (51773853) | y1 sc* v1 sev21; P{TRiP.HMS02302} attP2 (UAS-MICU1-RNAi) | BL419069 | |
| 10 000 Units/ml Penicillin and 10 mg/ml Streptomycin | Sigma Aldrich | Cat#P0781 | w,dTmbim5-KO;MICU1/Cyo | this paper | |
| L-Glutamine | Sigma Aldrich | Cat#G7513-100ML (53508901) | Oligonucleotides | | |
| Dimethyl sulfoxide | Sigma Aldrich | 276855-100 ML | Rp49_for:CTGTC GCACAAATGG CGCAA | this paper | |
| Trypsin/EDTA (0.05%/0.02%) | Sigma Aldrich | Cat#T3924-100ML (53508900) | Rp49_rev:GCGCTTG-TTCGATCCGTAAC | this paper | |
| Digitonin | Sigma Aldrich | Cat#D5628-1G | dTmbim5_for:AAC-CAGCATCGCG-GATAAT | this paper | |
| Chameleon Duo Pre-stained Protein Ladder | LiCor | Cat#928-60000 (51809546) | dTmbim5_re-v:AAAAGGCA-TAGGTGCTCTGG | this paper | |
| Native Marker Kit | GE Healthcare/ Thermo Fisher | Cat#GE17-0445-01 | dMCU_for:AGCACA-GAAGCTTCGCATC | this paper | |
| CGP-37157 | Sigma Aldrich | Cat#C8874-5MG | dMCU_rev:CACG-TATTCCACATA-GATGTCTTCTC | this paper | |
| Calcium Green-5N | Life Technologies | Cat#C3737 | dMICU1_for:GCTCA-CAGTCCTCTC-GATTTCACGACGC | this paper | |
| Thapsigargin | Life Technologies | Cat#T-7459 | dMICU1_rev:ACCCG-CAATGTGGTA-GAAAGTAAGGGC | this paper | |
| Polyethyleneglycol-3350 (PEG) | Merck Millipore | Cat#202398 | dMICU3_for:TTCG-CAACCTAAGG-GACAAG | this paper | |
| Experimental models: Cell lines | | | dMICU3_rev:-GAATCCT-GATTTCGGCTTTG | this paper | |
| HEK293 wildtype | 16 | | dEMRE_for:GAGC-CAT-CAAACCCAAGC | this paper | |
| HEK293-TMBIM5-Knockout | 16 | | dEMRE_rev:GGTAT-GACGGCACA-GAAGATG | this paper | |
| HEK293-MCU-Knockout | Gift of Kevin Foskett | | Recombinant DNA | | |
| HEK293-MICU1-Knockout | Gift of Kevin Foskett | | Plasmid:pCMV3-c-GFP-TMBIM5 | 16 | |
| HEK293-MICU2-Knockout | Gift of Kevin Foskett | | Plasmid:pCMV3-c-GFPspark | 16 | |
| HEK293-EMRE-Knockout | Gift of Kevin Foskett | | Plasmid:pCMV3-TMBIM5 D294R-D325N-GFP | 16 | |
| HAP1-wildtype | 18 | | | | |
| HAP1-TMBIM5-Knockout | 18 | | | | |
| Experimental models: Organisms/strains | | | | | |
| w[1118] fly (control fly) | this paper | | | | |
| w[1118],dTmbim5-KO/ (FM7) (Tmbim5 knockout fly) | this paper | | | | |
| w;MCU[1]/(TM6B) (MCU knockout fly) | 28 | | | | |
| w;MICU1[32]/ Cyo.GFP (MICU1 knockout fly) | 28 | | | | |
| w;MICU3[27]/(TM6B) (MICU3 knockout fly) | 28 | | | | |

| Software and algorithms | | |
|---|---|---|
| GraphPad Prism 10.2.3 | https://www.graphpad.com/ | RRID:SCR_002798 |
| AFFINITY Designer 2.5.7 | https://affinity.serif.com/de/designer/?srsltid=AfmBOopG_NxRhAscn7-SqpOzr-7jCRxqIUel3nOKKp9CG6g_z480WaH8 | RRID:SCR_016952x |
| Other | | |
| Trans-Blot 0.2 μm Nitrocellulose Mini | Bio-Rad | 170-4158 |
| Trans-Blot Turbo 0.2 μm Nitrocellulose Midi | Bio-Rad | 170-4159 |
| 4–20% Mini-PROTEAN® TGX Stain-Free™ Protein Gels, 15 well, 15 μl | Bio-Rad | #4568096 |
| 10x Tris/Glycine/SDS (5 l) (51776043) | Bio-Rad | 161-0772 |
| ATP Bioluminescence Assay Kit HS II | Merck | 11699709001 |
| Quick-gDNA Miniprep Kit | ZYMO RESEARCH | D3025 |
| Faststart Universal SYBR Green Master Kit | Sigma-Aldrich | 493850001 |
| BC Assay | Interchim | UP40840A (51798043) |
| High Capacity cDNA Reverse Transcription Kit | Applied Biosystems™, Thermo Fisher Scientific™ | 4368814 (51773689) |
| RNA MiniPrep Kit | Zymo Research | D4020 |

## Reporting summary

Further information on research design is available in the Nature Portfolio Reporting Summary linked to this article.

## Data availability

All data and uncropped immunoblots are contained in the Supplementary data file. No data was deposited. Requests for further information and resources should be directed to and will be fulfilled by the lead contact, Axel Methner (axel.methner@gmail.com). All unique/stable reagents generated in this study are available from the lead contact without restriction.

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

## Acknowledgements

This work was funded by the Deutsche Forschungsgemeinschaft (DFG) to AM (ME1922/17-1). WFG is grateful to the Austrian Science Fund (FWF) for Excellence Cluster 10.55776/COE14. We thank Marion Silies for providing lab space. The SIM equipment is part of the Nikon Center of Excellence, Graz, and is supported by the Austrian infrastructure program 2013/ 2014, Nikon Austria Inc., and BioTechMed.

## Author contributions

L.Z., B.G., A.K., F.D., S.B., D.B., and L.R.C. performed experiments and analyzed data. L.Z., B.G., and A.M. designed the experiments. V.G. and W.F.G. supervised and contributed ideas. L.Z., B.G., V.G., and W.F.G. edited the manuscript. A.M. conceptualized the study, obtained funding, and wrote the manuscript.

## Funding

## Competing interests

The authors declare no competing interests
