## [Transparent Peer Review file · Communications Biology]

Interaction of the mitochondrial calcium/proton exchanger TMBIM5 with MICU1

Corresponding Author: Professor Axel Methner

Version 0:

Reviewer comments:

Reviewer #1

(Remarks to the Author)

The present study is interesting, investigating the potential crosstalk between TMBIM5 and MCUC proteins. However, several areas require clarification or improvement, as outlined below:

It is well established that TMBIM5 knockout (T5 KO) impairs mitochondrial homeostasis. Please clarify the novelty of the current study in this context.

The conclusion that TMBIM5 acts upstream of MICU1 is primarily based on its rescue capability. However, this does not establish causality or signal directionality. The observed rescue could result from functional redundancy or buffering rather than a true hierarchical relationship. Additional data, such as time-course experiments, would strengthen the claim of upstream/downstream interactions.

In T5 KO, MCU expression appears unchanged at the protein level (Figure 1C). This suggests a unidirectional effect of MCU or its component knockouts on TMBIM5.

What could explain the lack of change in MICU2 levels in T5 KO, or vice versa? Does MICU2 play a redundant or distinct role compared to MICU1?

What is the age of the flies used for all molecular downstream analyses? Please provide a rationale for the chosen age.

Figure 2E: Please specify whether the ATP levels refer to total cellular ATP or mitochondrial ATP.

The potential reasons for reduced ATP levels should be discussed in relation to the observed mitochondrial phenotypes.

Figure 2F: The EM images are of poor quality. Please provide higher-resolution images, with improved focus or zoom on the cristae structure.

The authors state that the cristae structure is severely disrupted. However, it is unclear whether the cristae are broken, irregular, or show reduced inter/intra-cristae distance. Please elaborate on these structural alterations.

Please update reference #17 to its peer-reviewed published version in EMBO Journal.

Reviewer #2

(Remarks to the Author)

Zhang and colleagues present interesting data, novel reagents and insights into the molecular function of TMBIM5 homologues, and describe a functional relationship with MICU1. Previous work has established TMBIM5 as a regulator of mitochondrial $\text{Ca}^{2+}/\text{H}^{+}$ transport, and MICU1 is generally considered a key component of the mitochondrial calcium uniporter complex (MCUC) but has other emerging roles.

First, they study reciprocal effects on steady state protein levels of TMBIM5 and MCUC components with relatively modest

effects but TMBIM5 being upregulated upon loss of MCU or MICU1. Then they generate and validate a novel, targeted genetic knockout for fly *Tmbim5* and show impacts on organism fitness and survival, as well as multiple aspects of mitochondrial function including swelling and reduced calcium buffering capacity. A crucial aspect of the study reveals compelling genetic interaction with MCUC components, notably that partial loss of MICU1 ameliorates *dTmbim5* phenotypes. Exploring the interaction further, they find evidence of a macromolecular complex and similar impacts of LOF on IMM features. Finally, complementation assays assessing epistasis where TMBIM5 and MICU1 show non-reciprocal effects. Overall, the results reported here are from well-executed and well-controlled experiments, benefiting from multiple model systems, they provide novel insights and are logically interpreted. As such, I have only a few points to consider for revision.

Main points:

Fig. 1D. Is there an increase in TMBIM5 in EMRE KO cells? It seems incongruous that these blots are not quantified. Is LETM1 used as a loading control here? Why not Actin like the other blots?

Regarding the description of the new *dTmbim5* KO flies: this is a very welcome new tool and the generation and validation looks good, but the viability of the new KO line differs significantly from the pupal lethality the authors described previously (and mentioned here) for available knockdown lines. The caveats of RNAi approaches are well-understood, but I think the authors should make some comment on this, at least so it doesn't give the appearance that they haven't noticed or don't think it is important (it is, otherwise why bother going the extra mile to make the KO). Also, typical for validating phenotypes of new mutants, have the authors tried to rescue any of the key KO phenotypes by re-expressing wt *dTmbim5*?

Fig. 3. A. The median survival of *dTmbim5* KO flies differs substantially from that shown in Fig. 2D which appears to be about 10-15 days. Please clarify.

B. "Climbing speed". A measure of climbing speed is not described in methods and this description differs from that shown in 2C. Please clarify. Would 'climbing ability' be a sufficient descriptor?

What was the *Micu1* RNAi line? I didn't see this described in Methods.

E. It is a little incongruous that they show just images for TEM in 2F and just quantification in 3E. Some unity would be welcome. At least, it would be nice to see images for 3E.

As this is, as far as I know, the first description of genetic loss of *dTmbim5* and thus a valuable resource, have the authors been able to conduct and rescue experiments for any of the key phenotypes?

Regarding the differences in sub-mitochondrial localisation and effects on IMM features, have the authors been able to detect interactions of TMBIM5 and MICU1 with MIC60 or CHCHD2? It would be interesting to see if the relative binding changes upon loss of one or the other protein as they suggest.

Minor points:

- p3. For correctness, what is written as F0(zero)F1-ATP synthase should be FO(letter O for 'oligomycin-sensitive').
- Reference 17, a bioRxiv preprint, is now published.
- Fig. 4D. What do the vertical lines represent?
- P11. What does "MICU1 results in more prominent mitochondria" mean here?
- "*Drosophila melanogaster*" should be italicised, on p5 and p12.
- Discussion, second paragraph. Other hypotheses are conceivable. The authors might prefer to say 'we envisage two...'
- for future consideration, if the authors wish to measure mitochondrial mass via nucleic acids, mRNA is much less reliable than DNA (expression can be altered far easier than DNA content), although here they have another reliable measure with citrate synthase.

Version 1:

Reviewer comments:

Reviewer #1

(Remarks to the Author)

All of my concerns have been resolved by the authors.

Reviewer #2

(Remarks to the Author)

The authors have responded to my initial critiques adequately and the changes have improved the manuscript (in my opinion). I have no further critiques except to point out that the changes to Figure 3 are welcome but the in-text references to panels C-E now do not match the figure panels.

Also, the authors should check if the TEM images in 3C match the genotypes, or if they are truly representative: the *dT5*-KO looks fine while the *dT5*-KO; *MICU1*-KO/+ looks worse.

I look forward to seeing if RMBIM5, MICU1, CHCHD2 and MIC60 exist within a common macromolecular complex

Point-to-Point Response for COMMSBIO-25-3750-T

Reviewer #1 (Remarks to the Author)

Comment 1: Novelty of the study

Reviewer: "It is well established that TMBIM5 knockout (T5 KO) impairs mitochondrial homeostasis. Please clarify the novelty of the current study in this context."

Response: This is actually the first full animal knockout of TMBIM5. In our previous publication, we described the effect of a channel pore mutant (D326R) in mice. This point mutation results in downregulation of TMBIM5 and a possible loss of function ¹. In the present study, we now generated a full Tmbim5 KO in Drosophila. This model largely recapitulates the phenotype found in mice with a channel-pore mutation in TMBIM5, suggesting that the findings in mice are indeed caused by TMBIM5 loss of function. This novel fly model also allowed us to investigate genetic interactions with other proteins of the MCU complex, namely MICU1, MICU3, MCU and EMRE, which would be very hard to do in mice. These experiments revealed a genetic interaction with MICU1 which we explored further in human cell lines. This is completely novel. This has now been described more clearly in the revised introduction.

Comment 2: Upstream/downstream relationship claims

Reviewer: "The conclusion that TMBIM5 acts upstream of MICU1 is primarily based on its rescue capability. However, this does not establish causality or signal directionality. The observed rescue could result from functional redundancy or buffering rather than a true hierarchical relationship. Additional data, such as time-course experiments, would strengthen the claim of upstream/downstream interactions."

Response: We thank the reviewer for raising this important point. The reviewer is correct that rescue capability alone does not definitively establish a hierarchical upstream/downstream relationship. We acknowledge that the observed rescue could indeed result from functional redundancy, buffering effects, or compensatory mechanisms rather than a direct linear pathway. Regarding time-course experiments, we considered a sequential knockdown experiment but believe that the interdependence we observe between these proteins makes it challenging to dissect temporal sequences, as perturbation of either protein might rapidly affect the other's localization and function. We have revised our discussion to better reflect this more nuanced interpretation of the data and to avoid overstating claims about hierarchical relationships.

Comment 3: MCU expression levels

Reviewer: "In T5 KO, MCU expression appears unchanged at the protein level (Figure 1C). This suggests a unidirectional effect of MCU or its component knockouts on TMBIM5."

Response: MCU expression in Figure 1C is actually significantly downregulated. We removed the word "slight" to make this more obvious.

Comment 4: MICU2 vs MICU1 functions

Reviewer: "What could explain the lack of change in MICU2 levels in T5 KO, or vice versa? Does MICU2 play a redundant or distinct role compared to MICU1?"

Response: In the cited studies on the role of MICU1 with the cristae forming machinery and CHCHD2 interaction ^{2,3}, MICU2 was not studied. Only MICU1 but not MICU2 depletion protects against cold-induced ferroptosis ⁴. It is therefore probable that MICU2 does not share the non-MCUC functions of MICU1, which probably explains the lack of regulation.

In our view, the function of MICU2 is essentially dependent on MICU1. This is supported by observations that MICU2 is a regulator of MICU1 and not able to interact with the MCU complex itself. In addition, MICU2 cannot homo-oligomerize and MICU2 dimers are not able to bind to MCU. Only MICU1-MICU2 heterodimers are engaging the MCUC ⁵. So far only the KO of MICU1 was reported to influence the expression of MICU2 negatively ⁶. This has now been discussed in more detail in the revised manuscript.

Comment 5: Fly age for analyses

Reviewer: "What is the age of the flies used for all molecular downstream analyses? Please provide a rationale for the chosen age."

Response: All flies used for molecular downstream analyses were 4–7 days old adults. This age range was chosen because no fly lethality is observed during this period, and we aimed to avoid analyzing escapers. The age of flies is now indicated in the revised manuscript.

Comment 6: ATP measurement clarification

Reviewer: "Figure 2E: Please specify whether the ATP levels refer to total cellular ATP or mitochondrial ATP."

Response: We quantified total ATP levels of whole adult flies normalized to protein content.

Comment 7: ATP reduction discussion

Reviewer: "The potential reasons for reduced ATP levels should be discussed in relation to the observed mitochondrial phenotypes."

Response: We added the following to the results section: "Based on these results, we consider the reduced total ATP levels to be caused by compromised mitochondrial function (disrupted cristae architecture, mitochondrial swelling, and enhanced mPTP susceptibility) that manifests right before the drop in lifespan." This has now been addressed more clearly in the revised manuscript.

Comment 8: EM image quality

Reviewer: "Figure 2F: The EM images are of poor quality. Please provide higher-resolution images, with improved focus or zoom on the cristae structure."

Response: We have now provided higher-resolution images demonstrating changes in the cristae structure.

Comment 9: Cristae disruption details

Reviewer: "The authors state that the cristae structure is severely disrupted. However, it is unclear whether the cristae are broken, irregular, or show reduced inter/intra-cristae distance. Please elaborate on these structural alterations."

Response: We acknowledge that our transmission electron microscopy equipment has limitations for detailed cristae structural analysis. The specific cristae alterations in TMBIM5 deficiency have been extensively characterized in previous publications. Oka et al. (2008) demonstrated that TMBIM5 knockdown results in reduced cristae number, curved cristae morphology, and ring-like inner membrane structures. We have now added the following text to the manuscript to clarify this point: "While our transmission electron microscopy equipment limited detailed cristae structural analysis, previous studies have characterized these defects in detail, showing reduced cristae number, curved morphology, and ring-like inner membrane structures in TMBIM5-deficient cells ⁷."

Comment 10: Reference correction

Reviewer: "Please update reference #17 to its peer-reviewed published version in EMBO Journal."

Response: We thank the reviewer for pointing this out to us. This has now been corrected.

Reviewer #2 (Remarks to the Author)

Overall Assessment

Reviewer: "Zhang and colleagues present interesting data, novel reagents and insights into the molecular function of TMBIM5 homologues, and describe a functional relationship with MICU1... Overall, the results reported here are from well-executed and well-controlled experiments, benefiting from multiple model systems, they provide novel insights and are logically interpreted."

Main Points

Comment 1: Figure 1D quantification and controls

Reviewer: "Fig. 1D. Is there an increase in TMBIM5 in EMRE KO cells? It seems incongruous that these blots are not quantified. Is LETM1 used as a loading control here? Why not Actin like the other blots?"

Response: We appreciate the reviewer's questions regarding Figure 1D. We have revised the figure and now provide quantification of the relevant TMBIM5 KO lanes in new Figure 1D'.

The two rightmost lanes represent EMRE KO cells, which were intentionally loaded at a significantly higher total protein concentration compared to the TMBIM5 KO and WT samples. This was done specifically to assess the sensitivity of our EMRE antibody and confirm the complete absence of EMRE in the knockout line, serving as a robust negative

control for the antibody. This differential loading explains the seemingly higher signal for TMBIM5 in those lanes, as can also be observed in LETM1.

These experiments were conducted by the team of Vivek Garg, who still had some of the to our knowledge only functional, now discontinued antibody against EMRE. Vivek favors LETM1 as loading control over Actin (a cytosolic and cytoskeletal protein) because it allows adjustment for fluctuations in cellular mitochondrial content that could lead to inaccurate normalization and potentially false positive or negative results. The Garg team has extensively validated the stability of LETM1 protein levels across a range of genetic manipulations such as MCU KO, EMRE KO, and MICU1 KO cells. This consistent expression under various knockout conditions, particularly those involving mitochondrial proteins, makes it a highly reliable loading control for our studies.

In addition, we have previously reported that LETM1 protein levels are similar compared to beta-tubulin levels between wild-type and TMBIM5 KO cell lines (Zhang et al. 2022). We have now added new experiments comparing LETM1 levels in TMBIM5 KO and WT cells using Actin as loading control. These new data are shown in new Figure 1C.

Comment 2: dTmbim5 KO fly viability vs RNAi lethality

Reviewer: "Regarding the description of the new dTmbim5 KO flies: this is a very welcome new tool and the generation and validation looks good, but the viability of the new KO line differs significantly from the pupal lethality the authors described previously (and mentioned here) for available knockdown lines... Also, typical for validating phenotypes of new mutants, have the authors tried to rescue any of the key KO phenotypes by re-expressing wt dTmbim5?"

Response: We appreciate the reviewer's point regarding the phenotypic differences. Genetic knockouts can trigger compensatory mechanisms and developmental adaptation that are not possible with RNAi approaches, often resulting in milder phenotypes. Additionally, RNAi can have off-target effects and create dominant-negative scenarios that may be more detrimental than complete gene absence. We generated the KO line specifically to enable stable genetic studies and facilitate complex genetic crosses, which is not feasible with RNAi stocks.

The following has been added to the revised manuscript for clarification: "In contrast to RNAi-mediated knockdown of Tmbim5, Tmbim5 KO flies were viable. Such differences are often observed and usually attributed to the fact that complete knockouts allow developmental adaptation and compensatory gene expression that can mitigate phenotypic severity ⁸."

Rescue experiments using the mild ubiquitous driver *daughterless*-Gal4 > UAS-*dTmbim5* confirmed that the lifespan defect was specifically due to *Tmbim5* loss. These data have been added to the new Figure 2D. We observed that overexpression of dTmbim5 with strong ubiquitous drivers such as Actin-Gal4 or Tubulin-Gal4 is lethal (not shown).

Comment 3: Survival discrepancy between figures

Reviewer: "Fig. 3. A. The median survival of dTmbim5 KO flies differs substantially from that shown in Fig. 2D which appears to be about 10-15 days. Please clarify."

Response: We also noticed the discrepancy in median survival between dTmbim5 KO flies in Fig. 2D and Fig. 3A. The flies in Fig. 2D represent early generations immediately after establishing the knockout, whereas those in Fig. 3A are from a stock maintained for approximately one year. We observed improved survival over successive generations, likely reflecting genetic compensation or suppressor mutations that commonly arise in laboratory stocks over time. This phenomenon has been well-documented in Drosophila knockout lines and does not affect the genetic interaction conclusions of our study.

Comment 4: Climbing assay terminology

Reviewer: "B. 'Climbing speed'. A measure of climbing speed is not described in methods and this description differs from that shown in 2C. Please clarify. Would 'climbing ability' be a sufficient descriptor?"

Response: We thank the reviewer for pointing this out. This has now been corrected to "climbing ability" in results, figures and figure captions. The missing Materials and Methods part has been added.

Comment 5: MICU1 RNAi line information

Reviewer: "What was the Micu1 RNAi line? I didn't see this described in Methods."

Response: We thank the reviewer for pointing this out. The information for the UAS-MICU1-RNAi line (Bloomington #419069) has been added to the Materials and Methods section.

Comment 6: TEM image consistency

Reviewer: "E. It is a little incongruous that they show just images for TEM in 2F and just quantification in 3E. Some unity would be welcome. At least, it would be nice to see images for 3E."

Response: We have now supplied exemplary images in new Figure 3C.

Comment 7: Rescue experiments for dTmbim5 KO

Reviewer: "As this is, as far as I know, the first description of genetic loss of dTmbim5 and thus a valuable resource, have the authors been able to conduct and rescue experiments for any of the key phenotypes?"

Response: We conducted the rescue experiments with flies with partial Micu1 depletion as shown in Figure 3, which demonstrate the key genetic interactions that are the focus of this study. While we have initiated additional rescue experiments with other players, these studies are part of ongoing work that extends beyond the scope of the current manuscript and will be reported in future publications.

Comment 8: TMBIM5-MICU1 interactions with MIC60/CHCHD2

Reviewer: "Regarding the differences in sub-mitochondrial localisation and effects on IMM features, have the authors been able to detect interactions of TMBIM5 and MICU1 with MIC60 or CHCHD2? It would be interesting to see if the relative binding changes upon loss of one or the other protein as they suggest."

Response: We have not yet addressed this question but greatly thank the reviewer for this insightful suggestion. Indeed, these reciprocal localization dependencies combined with the shared interaction partners CHCHD2 and MIC60 support the hypothesis that MICU1, TMBIM5, CHCHD2, and MIC60 exist within a common macromolecular complex – a relationship that merits investigation in future studies.

Minor Points

Comment 9: ATP synthase nomenclature

Reviewer: "p3. For correctness, what is written as F0(zero)F1-ATP synthase should be FO(letter O for 'oligomycin-sensitive')."

Response: We thank the reviewer for pointing this out. This has now been corrected.

Comment 10: Reference update

Reviewer: "Reference 17, a bioRxiv preprint, is now published."

Response: We thank the reviewer for pointing this out. This has now been corrected.

Comment 11: Figure 4D clarification

Reviewer: "Fig. 4D. What do the vertical lines represent?"

Response: Vertical lines in 4D indicate the TMRM concentration threshold for half-maximal IBM association index changes. Lower thresholds reflect more even membrane potential distribution between cristae and inner boundary membrane; higher thresholds indicate greater potential differences between these compartments. Using this analysis, we observed a reduced TMRM concentration for half maximal changes in the IBM association index in TMBIM5 and MICU1 KO cells that were more pronounced in MICU1-deficient cells (Figure 4D). These data indicate that the knockdown of either TMBIM5 or MICU1 yields an equilibration of the membrane potential between cristae and IBM. This has been clarified in the revised manuscript.

Comment 12: Language clarification

Reviewer: "P11. What does 'MICU1 results in more prominent mitochondria' mean here?"

Response: We thank the reviewer for pointing this out. The sentence "In TMBIM5 KO cells, MICU1 results in more prominent mitochondria despite lower association with the IBM." (page 11; line 11-12) was now changed to "In TMBIM5 KO cells, MICU1 overexpression results in more branched mitochondria despite lower association with the IBM." We apologize for the mistake.

Comment 13: Formatting corrections

Reviewer: "Drosophila melanogaster should be italicised, on p5 and p12."

Response: We thank the reviewer for pointing this out. This has now been corrected.

Comment 14: Discussion language

Reviewer: "Discussion, second paragraph. Other hypotheses are conceivable. The authors might prefer to say 'we envisage two...'"

Response: We thank the reviewer for pointing this out. This has now been corrected.

Comment 15: Methodological note

Reviewer: "For future consideration, if the authors wish to measure mitochondrial mass via nucleic acids, mRNA is much less reliable than DNA (expression can be altered far easier than DNA content), although here they have another reliable measure with citrate synthase."

Response: We thank the reviewer for pointing this out. We actually measured mitochondrial content using DNA and not mRNA. The mention of mRNA in the main text was a mistake. The Materials and Methods section accurately describes the quantification of mitochondrial DNA (mtDNA) relative to nuclear DNA using qPCR (see Methods: "Quantification of mitochondrial and nuclear DNA"). We have corrected the text.

References

1. Zhang, L. *et al.* TMBIM5 loss of function alters mitochondrial matrix ion homeostasis and causes a skeletal myopathy. *Life Sci. Alliance* **5**, e202201478 (2022).
2. Gottschalk, B. *et al.* MICU1 controls cristae junction and spatially anchors mitochondrial Ca²⁺ uniporter complex. *Nat. Commun.* **10**, 3732 (2019).
3. Tomar, D. *et al.* MICU1 regulates mitochondrial cristae structure and function independently of the mitochondrial Ca²⁺ uniporter channel. *Sci. Signal.* **16**, eabi8948 (2023).
4. Nakamura, T. *et al.* The mitochondrial Ca²⁺ uptake regulator, MICU1, is involved in cold stress-induced ferroptosis. *EMBO Rep.* e51532 (2021).
5. Patron, M. *et al.* MICU1 and MICU2 finely tune the mitochondrial Ca²⁺ uniporter by exerting opposite effects on MCU activity. *Mol. Cell* **53**, 726–737 (2014).
6. Plovanich, M. *et al.* MICU2, a paralog of MICU1, resides within the mitochondrial uniporter complex to regulate calcium handling. *PLoS One* **8**, e55785 (2013).
7. Oka, T. *et al.* Identification of a novel protein MICS1 that is involved in maintenance of mitochondrial morphology and apoptotic release of cytochrome c. *Mol. Biol. Cell* **19**, 2597–2608 (2008).
8. Rossi, A. *et al.* Genetic compensation induced by deleterious mutations but not gene knockdowns. *Nature* **524**, 230–233 (2015).